# Comparative Analysis of the Antioxidant and Anti-Inflammatory Effects of Krill and Fish Oil

**DOI:** 10.3390/ijms26157360

**Published:** 2025-07-30

**Authors:** Esra Tansu Sarıyer, Murat Baş, Meral Yüksel

**Affiliations:** 1Department of Nutrition and Dietetics, Institute of Health Sciences, Acibadem Mehmet Ali Aydinlar University, 34752 Istanbul, Turkey; dytesrasariyer@gmail.com; 2Department of Nutrition and Dietetics, Faculty of Health Science, University of Health Sciences, 34668 Istanbul, Turkey; 3Department of Nutrition and Dietetics, Faculty of Health Sciences, Acibadem Mehmet Ali Aydinlar University, 34752 Istanbul, Turkey; 4Department of Medical Laboratory Techniques, Vocational School of Health-Related Services, Marmara University, 34865 Istanbul, Turkey; meralyuksel@marmara.edu.tr

**Keywords:** krill oil, fish oil, antioxidant, anti-inflammatory, long-chain polyunsaturated fatty acids

## Abstract

Krill oil (KO) and fish oil (FO) are rich sources of long-chain polyunsaturated fatty acids, with eicosapentaenoic acid (EPA) and docosahexaenoic acid (DHA) bound to distinct molecular carriers (phospholipids vs. triglycerides). These oils have been the subject of considerable research interest over the past few years owing to their roles extensively studied for their antioxidant and anti-inflammatory properties relevant to disease prevention and therapy in certain diseases. This review aimed to provide a comparative summary of the antioxidant and anti-inflammatory activities of KO and FO, based on their bioactive components, and highlight the similarities and differences in their prospective mechanisms of action. Both oils exert antioxidant and anti-inflammatory activities, aligning with the review focus. The bioactivities of both oils stem from their distinct molecular compositions: KO delivers EPA/DHA via phospholipids, alongside astaxanthin, while FO provides EPA/DHA bound to triglycerides. In some cases, they exhibit similar outcomes, whereas in others, one may be more effective than the other. Further comparative studies examining dose-dependent effects, bioavailability kinetics, and tissue-specific molecular pathways are warranted.

## 1. Introduction

Recent studies have reported the beneficial effects of omega-3 (n-3) long-chain polyunsaturated fatty acids (LC-PUFAs) on various health conditions, particularly cardiovascular diseases, obesity, skin diseases, arthritis, ulcerative colitis, ulcers, and inflammatory diseases [1,2,3]. Products derived from marine organisms, including fish oil (FO) and krill oil (KO), are the most essential dietary sources of LC-PUFAs. Moreover, KO and FO are notable for other bioactive components they contain (Figure 1) [1,2,3].

Krill are small, red-colored crustaceans found in the Arctic and Antarctic oceans. The largest krill biomass is observed in the Antarctic Ocean, reaching approximately 1 billion tons [2,3]. Despite their small size, reaching only 4.5–6 cm in length, krill are highly valued for their bioactive compound content. KO comes from the Antarctic species *Euphausia superba*, the largest of its kind. It contains 30% eicosapentaenoic acid (EPA) and docosahexaenoic acid (DHA), astaxanthin, sterols, vitamins E and D, and 40% phospholipids [2,3]. Astaxanthin (3,30-dihydroxy-b, b-carotene-4,40-dion), a strong antioxidant and xanthophyll carotenoid, has a molecular structure with hydroxyl and keto moieties on each ionone ring with high antioxidant properties. The basic skeleton is an unsaturated hydrocarbon chain of 40 carbon atoms with thirteen conjugated double bonds, symmetrical with respect to positions 15-15′. The polyene chain in the structure of astaxanthin inhibits radicals in the cell membrane, while the terminal ring scavenges radicals in the outer and inner parts of the cell membrane [4]. Although the astaxanthin content in krill oil varies, it ranges from 40 to 5000 mg/kg [2,5,6,7].

One of the richest dietary sources of LC-PUFAs (EPA and DHA) is FO obtained from fatty fish tissues. EPA and DHA are n-3 fatty acid (FA) derivatives. In recent years, interest in LC-PUFAs, particularly n-3 FAs, has been increasing due to their reported beneficial health effects and reduced risk of chronic health conditions, including cardiovascular diseases, cancer, obesity, and diabetes. As LC-PUFAs are not synthesized in the body, they are essential FAs and should be included in the diet. α-Linolenic acid (ALA), EPA, and DHA represent the main n-3 FAs. The precursor of EPA and DHA synthesis is ALA, which is elongated in the endoplasmic reticulum by the action of elongase, and the formation of double bonds is catalyzed by desaturases. As a result of the action of Δ5 and Δ6 desaturases and elongases, α-linolenic acid is converted first to EPA (C20:5, n-3), then to docosapentaenoic acid (DPA, C22:5, n-3), and finally to DHA (C22:6, n-3) [8]. ALA, a plant-derived n-3 FA, is abundant in flaxseed, rapeseed oil, and walnuts, whereas the most significant food source of EPA and DHA is the meat of oily fish, including sardines, herring, mackerel, menhaden, salmon, cod liver, and other seafood (seals and whales) [9,10,11].

Several studies have compared the content, bioavailability, and health effects of KO and FO. The DHA content of FO is comparable to that of KO; however, the EPA content is lower. The main differences between KO and FO are summarized in Figure 1. In KO, more than 80% of EPA and DHA are rich in phospholipids comprising phosphatidylcholine, and LC-PUFAs are in the form of phospholipids, whereas those in FO are in the form of triglycerides. Unlike FO, KO is rich in vitamin A and astaxanthin, a carotenoid with strong antioxidant activity [7,12].

Furthermore, KO is a source of vitamin E derivatives, with more than 90% of its tocopherols in the form of α-tocopherol, and therefore, it has a high antioxidant capacity [7,12]. Studies on the bioavailability of KO and FO have not reached a clear conclusion on which doses and which n-3 PUFA sources or formulations are more effective for absorption. However, it is believed that the bonding of omega-3 FAs with phosphatidylcholine in the structure of KO increases bioavailability, positively contributing to health [13]. Although some studies have shown no difference between KO and FO in terms of bioavailability [13,14], some have reported that KO has a higher bioavailability [1,7]. In a study by Schuchardt et al., the bioavailability of triglyceride-form FO, ethyl-ester-form FO, and KO was compared, and no significant difference was observed [15].

The differences in the chemical structures of KO and FO are important factors affecting the absorption, bioavailability, and tissue distribution of LC-PUFAs. FO formulations typically consist of triglycerides, emulsions, re-esterified triacylglycerols, ethyl esters, and microencapsulated powders. In contrast, KO contains EPA and DHA in the PL form, which includes both high and low PL formulations, as well as phospholipid/free FA combinations. The digestion of triglycerides begins in the stomach, where lingual and gastric lipases hydrolyze them. However, only a small portion of PLs serve as substrates for these enzymes, and the majority of PLs are hydrolyzed in the small intestine by pancreatic phospholipase A2 with the assistance of other lipases [1,16,17]. PLs possess emulsifying properties that enhance the lipase’s ability to break down lipids, thereby improving lipid digestion and absorption. PLs can be absorbed in their undigested form or as lysophosphatidylcholine after digestion by enzymes in the small intestine. Unlike the hydrolysis products of TG, the relevant products from PLs do not depend on bile salts to form micelles. After being taken up by enterocytes, PLs are primarily incorporated into the surface layer of chylomicrons, while TGs dissolve in the interior of chylomicrons. Studies suggest that a significant portion of the dietary PL fraction is integrated into particles already present in the intestine and subsequently incorporated into the plasma HDL pool. It has been proposed that FA absorption in the small intestine is equivalent for both PL- and TG-derived compounds [1,16,17].

A recent network meta-analysis demonstrated that KO in PL form demonstrates superior absorption and efficacy at lower doses compared to FO in triglyceride form. High-dose FO (over 3000 mg, up to 4400 mg of DHA + EPA per day) was most effective in increasing Omega-3 Index (O3i), while KO at doses below 2000 mg showed better absorption. The KO PL form was found to be more effective in increasing area under the curve (AUC), while the FO emulsion form was more effective for area under the curve (Cmax) [17]. In another study, the administration of a single dose of EPA and DHA as re-esterified TAG, ethyl esters, or marine PLs to healthy individuals resulted in an increase in the O3i with the following efficacy: PL > TAG > ethyl esters [13].

The absorption and metabolic processing of PL-DHA in the KO structure, particularly in the nervous and hepatic systems, is notable. This selective mechanism forms a critical pathway for brain DHA accumulation, which is particularly important during neurodevelopment. The transport of DHA in the PL form to the brain occurs via the MFSD2A transporter protein, which is located in the blood–brain barrier and recognizes only the lysophosphatidylcholine-DHA (LPC-DHA) form [16]. Many studies indicate that the absorption of DHA in the PL form, particularly in the brain, is higher in KO [18,19,20]. Additionally, astaxanthin in KO increases the oxidative stability of the oil, reducing the degradation of EPA/DHA. In terms of elimination, both forms of EPA and DHA are metabolized in the liver via β-oxidation and then excreted as carbon dioxide and water. Due to its PL structure, it is suggested that KO has a more stable structure and remains in tissues for a longer period of time [1,6,17,21].

KO and/or FO products obtained by different methods are sold commercially and may differ in terms of bioavailability. KO and FO are extracted and processed using various methods. KO is obtained from fresh or dried krill, but fresh krill must be processed quickly after being caught. This is because the shells contain proteolytic enzymes that rapidly induce autolysis. KO extraction methods include solvent-based and solvent-free extractions, supercritical and subcritical fluid extractions, and enzymatic extractions [22]. FO extraction methods also vary between traditional and more novel methods. Traditional methods are used at an industrial level and consume a significant amount of energy and solvents (such as wet pressing and extraction using solvents or heat). New, environmentally friendly, and sustainable methods like supercritical carbon dioxide extraction, ultrasound-assisted extraction, and enzymatic hydrolysis offer more advantages compared to traditional methods [23].

The most commonly used traditional method for KO extraction is solvent extraction, which typically involves the use of solvents such as ethanol, isopropanol, acetone, and n-hexane. Although this method is the simplest and most economical, the environmental impact is higher. It is also time-consuming due to its multiple steps. A combination of polar and non-polar solvents is important for optimal extraction and to prevent damage to minor components [5,7,24]. Alcohols such as isopropanol and ethanol can extract large amounts of PLs from krill meal but produce krill oil with low levels of minor components. Acetone, while effective for extracting minor components, cannot extract PLs. Hexane appears to be the most commonly used solvent in vegetable oil production due to its relatively low cost, high extraction efficiency, and low residue levels in the meal. However, it has only moderate effectiveness in extracting both PLs and minor components from krill [5,7,24]. Extraction conditions also varied; Yin et al. performed extraction at 9 mL/g ethanol, 50 °C for 90 min, while Sun et al. used ethanol–hexane (4:6 *v*/*v*), 40 °C for 4 h, and Xie et al. used krill meal–acetone 1:2 *w*/*v* in the first stage; 4 °C for 15 min, krill meal–hexane 1:2 *w*/*v* at 30 °C for 15 min in the second stage, and krill meal–ethanol 1:3 *w*/*v* at 30 °C for 20 min in the third stage [25,26,27]. When ethanol–hexane (4:6, *v*/*v*) single-stage extraction was used to extract Antarctic KO from wet-based Antarctic krill directly, the highest yield of 97.72%, higher astaxanthin yield, and PUFA content were obtained, and it was observed that the active components were effectively preserved [28]. This method is also frequently used in FO extraction. Similarly, oils are dissolved using organic solvents such as n-hexane, ethanol, acetone, and chloroform [29].

Another method used is aqueous enzymatic extraction, in which krill cell walls are broken down using endogenous or exogenous enzymes to extract oil. The main enzymes used are trypsin-like serine protease, chymotrypsin-like serine protease, carboxypeptidase A enzyme, carboxypeptidase B enzyme, glucanase, and chitinolytic enzyme [5,24]. In the study conducted using the aqueous enzymatic extraction method, Antarctic krill was crushed and dispersed in 30% (*w*/*w*) water, with soybean oil used as a co-solvent. After 6 h of autolysis at pH 7.5 and 43.2 °C, the yields of Antarctic KO and astaxanthin were 76.7% and 44.24 mg/kg, respectively [30]. Although this method appears to be quite effective in simultaneously separating protein and krill peptides while preserving nutritional value, it has several disadvantages, including high water consumption, high cost, lengthy reaction times, and the risk that environmental conditions, such as temperature and pH, may affect the enzymes. Additionally, emulsion formation can also complicate the separation of oils [5,7,24]. Supercritical and subcritical extraction, a solvent-free technique, is an environmentally friendly method because it utilizes supercritical carbon dioxide, which is safe, non-toxic, and chemically inert. This method yields high-quality KO that is resistant to oxidation. However, carbon dioxide is not an ideal molecule for extracting all lipid content from krill because it cannot extract PLs, resulting in low phospholipid yield [7]. However, it has been observed that adding 5–20% ethanol to the extraction process can improve the solubility of PLs and thus enhance lipid recovery [31]. Compared to the traditional method, the content of astaxanthin and tocopherol is also better preserved [26]. Extraction of krill oil using subcritical butane at 30 °C and 0.3–0.8 MPa has yielded similar results in terms of yield and oil quality to hexane extraction, but it is faster and requires less solvent [32]. However, the equipment is expensive and has limited production capacity, making industrial production challenging [7]. The three-liquid-phase salting-out extraction system (TLP-SOES) consists of salts and organic solvents for the separation of oil, protein, and polysaccharides. These components are distributed into different phases according to polarity (oil-soluble, alcohol-soluble, water-soluble) and solubility. The advantages of this method include the ability to use wet krill, eliminating the need for drying, its lowest cost, easy scalability, the highest product recovery, lower viscosity, and the ability to form different phases in a short time [5,7,24]. The TLP-SOES extraction of Antarctic KO yielded three phases: a lipid phase, a protein-polysaccharide phase, and an aqueous phase (salt phase). It was found that ethanol/n-hexane solvent is effective, while acetone, butanol (t-butanol may be toxic), and propanol are not effective. However, as the ethanol concentration increases, the oil mixes with ethanol, and the yield decreases. Additionally, high salt usage may reduce DHA content [5,7,24]. In the study conducted using the TLP-SOES method, the following conditions were set: 16.5% ammonium citrate salt, 17.5% ethanol, 46% n-hexane, and a mixing time of 70 min. Under these conditions, a yield of 91.85% from KO, >90% DHA and EPA, 88.34% protein, and 79.67% isoflavone yield was achieved [33].

The traditional, or physical, method of FO extraction involves wet or dry cooking and pressing, followed by filtration and the removal of solid proteins, which become fish meal. It is generally used to obtain visceral fat. Most of the fish oil produced is used in seafood products, but a small portion is refined through neutralization, degumming, bleaching, and deodorization to obtain high-quality oil suitable for human consumption and the pharmaceutical industry. Although it is a simple and solvent-free method, there is a high risk of oxidation due to thermal processing [22,23,34].

Another method is the Soxhlet extraction, which involves continuous extraction using organic solvents such as alcohol, n-hexane, and diethyl ether. The Soxhlet technique is a continuous solid–liquid extraction process, which means the material to be extracted is in a solid mixture form, and the process is repeated until completion. This method has high efficiency and easier temperature control. However, the risks of solvent residue and high energy consumption are its disadvantages. In the ultrasound-assisted extraction (UAE) method, high-frequency sound waves are used to disrupt cell structure and release oil. Although it provides high yield and high-quality oil in a short time, it may not be frequently preferred due to the high cost of equipment. Although it appears to be a highly advantageous method, like all other methods, it is important to remember that environmental factors such as geographical region and season affect the efficiency of both KO and FO extraction [22,23,35].

This review aimed to compare the antioxidant and anti-inflammatory effects of KO and FO, which are believed to exert beneficial effects due to their bioactive components, and to demonstrate the similarities and differences in their possible mechanisms of action. Despite chemical compositional differences, systematic comparisons of their effects on inflammatory/oxidation pathways remain limited. Details of our search are provided in the Section 4. In this context, in vivo, in vitro, animal, and human studies investigating the effects of KO and FO, as well as examining inflammation and oxidative stress biomarkers, were included up to 2025. A summary of the literature is presented in the text.

## 2. Comparison of Anti-Inflammatory Effects of FO and KO

Several studies have indicated that KO regulates the release of pro-inflammatory cytokines, including tumor necrosis factor-alpha (TNF-α), interleukin-1 beta (IL-1β), and IL-6, and exerts anti-inflammatory effects by increasing IL-10 expression [2,36]. In lipopolysaccharide (LPS)-induced inflammation in the human acute monocytic leukemia cell line THP-1, KO–water emulsion treatment inhibited TNF-α release by inactivating macrophage binding to toll-like receptor-4 (TLR4), which plays a significant role in the innate immune response. In this in vitro study, when the responses depending on the KO doses applied were investigated, LPS inhibited TLR4 binding by 50% at 12.5 µg/mL and 75% at 25 µg/mL, whereas at 50 µg/mL, LPS completely eliminated TLR4 binding [37]. Researchers have proposed two possible mechanisms to explain this effect of KO. The first one is that the phospholipids in KO bind and neutralize LPS endotoxin, and the second one is the anti-inflammatory effect of omega-3 FA in its structure [37].

Another in vitro human macrophage study revealed that KO dose-dependently reduced LPS-induced IL-1β and TNF-α expression by regulating a broad spectrum of signaling pathways, including the stimulation of nuclear factor kappa B (NF-κB) and nucleotide binding and oligomerization domain (NOD)-like receptor [38]. NF-κB represents a family of inducible transcription factors that regulate several genes involved in different immune and inflammatory response processes, and dysregulated activation of this pathway contributes to the pathogenic processes of various inflammatory diseases [39].

Furthermore, human studies have supported the anti-inflammatory effect of KO. Lobraico et al., conducted a randomized double-blind controlled crossover study in which individuals with type 2 diabetes were randomized to KO (1000 mg/day) or olive oil supplementation for 4 weeks, and after a 2-week washout period, they switched to the other supplement for 4 weeks. All participants were subsequently offered an additional 17-week KO supplementation. Results revealed improved endothelial function and decreased C-peptide levels and HOMA scores in those receiving KO supplementation compared with those receiving olive oil [40]. A study conducted in individuals with chronic inflammation showed that 300 g/day KO supplementation significantly reduced serum C-reactive protein (CRP) levels within 7–14 days [41]. Berge et al. demonstrated that KO supplementation with 832.5 mg/day of EPA + DHA for 28 days did not change serum CRP levels but improved lipid profile in 17 healthy adults aged 18–36 years [42]. Stonehouse et al. investigated the effect of 1 g/day KO supplementation for 6 months in a multicenter double-blind randomized placebo-controlled study in 465 adults aged 40–65 years with mild-to-moderate osteoarthritis. KO use decreased knee pain score, whereas IL-6, TNF-α, and high-sensitive CRP (hs-CRP) levels remained the same [43]. Hence, KO’s anti-inflammatory activity arises synergistically from (i) phospholipid-bound EPA/DHA modulating eicosanoid pathways, (ii) astaxanthin inhibiting NF-κB/MAPK, and (iii) vitamin E stabilizing membranes [44,45].

For several years, the effects of FO on inflammation have been investigated. In a double-blind placebo-controlled randomized trial, Kiecolt-Glaser et al. noted that n-3 supplementation (2.5 g/day, EPA 2085 mg and DHA 348 mg) for 12 weeks reduced IL-6 and TNF-α production [45]. Another study showed that FO supplementation (1.8 g of EPA + DHA/day) for 26 weeks in individuals aged ≥65 years caused a decrease in the expression of genes involved in inflammatory and atherogenic-related pathways, including NF-κB signaling, eicosanoid synthesis, scavenger receptor activity, adipogenesis, and hypoxia signaling, on peripheral blood mononuclear cells [46]. A recent experimental study conducted on a mouse model of unilateral ureteral obstruction reported that n-3 supplementation (α-LA) inhibited macrophage chemotaxis and reduced the inflammatory response by decreasing monocyte chemotactic protein-1 (MCP-1) and CC motif chemokine receptor 2 expression [47]. LC-PUFAs primarily contributed to the beneficial effects of FO on inflammation. In plants, 15-desaturase aids in the synthesis of linoleic acid (LA) and ALA. In contrast, ALA synthesis does not occur in humans because the enzyme 15-desaturase is not present in the human body. Through a series of physiological processes in the body, ALA is converted to EPA and subsequently to DHA. Eicosanoids, such as PG, thromboxane, and leukotrienes, which are generated owing to the metabolism of these FAs, generally have anti-inflammatory effects [10]. Additionally, it is believed that n-3 FAs may have anti-inflammatory effects by inhibiting NF-κB activity, which plays a key role in inflammation, and increasing PPAR-γ and G protein-coupled receptor activities [48].

Studies investigating the superior effects of KO and FO on chronic and/or acute inflammation have yielded conflicting results. A study in an experimental arthritis model supplemented DBA/1 mice with 0.44 g/100 g of KO or FO for 68 days. KO decreased arthritis scores by 47.5% vs. FO’s 25.9% in mice. However, FO-fed mice with arthritis exhibited higher serum levels of IL-1α and IL-13 than those in the KO and control groups. The researchers suggested that FO may alter the cytokine milieu towards a more Th2 profile, and therefore, there may be an increase in IL-13 levels. They also criticized the study for suggesting that most of the cytokine data indicate that systemic cytokines are not a reliable biomarker for the severity of the disease. It was concluded that in future studies, it may be more appropriate to investigate cytokine mRNA expression directly within the joint [49]. In an in vivo study in experimental animals conducted to evaluate its anti-inflammatory and anti-analgesic effects, Neptune KO was shown to be more effective in suppressing pain and inflammation than FO [50]. In an experimental study in obese Zucker rats, the effects of KO- and FO-rich diet (0.5 g of EPA + DHA/100 g dietary intake) on ectopic fat and inflammation were investigated for 4 weeks. The KO-fed group exhibited lower liver and heart TAG concentrations than the FO-fed group, whereas both (n-3) LC-PUFA diets reduced plasma low-density lipoprotein-cholesterol levels and LPS-induced TNF-α release from peritoneal macrophages to the same extent. Furthermore, KO and FO treatment upregulated endocannabinoid concentrations in visceral adipose tissue, liver, and heart, which may affect lipid metabolism in these tissues via CB1 (type 1 cannabinoid receptor) and/or CB2 receptors [51]. In a double-blind randomized clinical trial by Cicero et al., individuals with moderate hypertriglyceridemia were administered 1000 mg of n-3 ethyl ester PUFA or 500 mg of KO supplement twice daily for 4 weeks. Participants were assigned to the alternative treatment for 4 weeks following a 4-week post-treatment washout period. KO demonstrated lipid-lowering effects (according to serum total triglycerides, LDL cholesterol, apolipoprotein-B, ALT, and AST levels) comparable to those achieved with a 4-fold higher dose of purified n-3 ethyl ester PUFAs and more effectively reduced hs-CRP [52]. In an experimental study in transgenic C57BL/6 hTNF-α mice, high-fat control diet (24.50% total fat, *wt*/*wt*) or high-fat diets containing similar FO or KO doses were administered for 6 weeks. Both fats reduced inflammation and improved lipid metabolism; although both were effective in reducing plasma triacylglycerol, KO was found to be more effective. Pro-inflammatory cytokine levels were not significantly different between the treatment groups, except for increased IL-17 levels in FO-fed mice and a trend toward decreased MCP-1 levels in the KO-fed group. FO-induced IL-17 elevation may reflect n-3 PUFA modulation of Th17 differentiation [53]. In a study by Tou et al., female Sprague-Dawley rats were fed a high-fat 12% (*wt*) diet containing corn oil or n-3 PUFA-rich flaxseed oil, krill, menhaden, salmon, or tuna oil for 8 weeks; the results showed no significant differences between the groups in terms of serum-2 PGs, thromboxane B metabolites, and oxidative stress biomarkers [54]. A study investigating the effects of KO, FO, and astaxanthin supplementation by oral gavage (2.5% *v*/*w*) for 4 weeks in an experimental ulcer model revealed no significant difference between the KO and FO groups in inflammatory biomarkers [55]. Another study investigating the effects of KO and FO on carrageenan-induced inflammation observed that both oils showed beneficial effects, with KO’s effects being superior in suppressing pain and IL-6 and TNF-α expression. The researchers proposed that the mechanism of action could be inhibiting cyclooxygenase (COX) synthesis, thereby reducing PG (PGF 2 α-EA in mice), contributing to the triggering of pain perception and nociceptive neuron hyperactivity in the dorsal horn [50].

n-3 PUFAs regulate the activation profile of microglia, resulting in anti-inflammatory or neuroprotective effects. This regulation has the potential to reduce neuroinflammation, restore neural function, and potentially prevent the progression or recurrence of depression [56]. LC PUFAs exhibit neuroprotective properties by reducing pro-inflammatory cytokines, decreasing infarct size, and improving cognition after severe, acute neurological damage such as hypoxic brain ischemia [57]. Studies investigating the effects of LC-PUFAs on neuroinflammation and neurological diseases have focused primarily on DHA. Unlike EPA, which is oxidized after entering the brain from plasma, unesterified DHA is largely (more than 80%) and selectively transported to the sn-2 position of membrane phospholipids, particularly ethanolamine glycerophospholipids and phosphatidylcholine, via an acyl-CoA synthetase and acyltransferase pathway. As a component of cell membranes, DHA contributes to membrane fluidity and is normally found in brain and retinal cells [58]. However, EPA is also one of the fatty acids that has been studied for neuroinflammation. In a study conducted on aged rats, after a 4-week EPA dietary supplement, microglia regulation, IL-4 elevation, and IL-1β suppression were observed in the hippocampus [59]. The effects of LC n-3 on diet-induced low-grade neuroinflammation have also been investigated with similar findings. In rats fed a high-fat diet (HFD) supplemented with FO, a decrease in hypothalamic microglial activity and the synthesis of pro-inflammatory cytokines TNF-α and IL-6 were observed [60,61]. In recent years, the role of KO, in addition to FO, in neurological diseases associated with inflammation and oxidative stress has been investigated. Du et al. reported that long-term (2 months) KO administration (50 and 100 mg kg^−1^ BW) to APP/PS1 mice improved memory abilities, reduced amyloid β concentration by regulating the amyloidogenic pathway, inhibited neuroinflammation by regulating the TLR4-NLRP3 signaling pathway, and prevented neuronal damage. Additionally, it was found to reduce early-stage mild cognitive impairment by altering the gut microbiota, thereby affecting the gut microbiota–short-chain fatty acids–brain axis [62]. Similarly, in an amyloid β-induced Alzheimer’s disease mouse model, supplementation with KO at different doses (100, 200, and 500 mg/kg/day) for 14 days resulted in a shorter time spent in the Morris maze test and improvements in learning and memory abilities. Additionally, it resulted in a significant decrease in oxidative stress parameters (ROS, MDA, and NO levels) compared to the control group and regulation of the apoptotic signaling pathway (the Bax/Bcl-2 ratio decreased, suppressing cell death). Another issue to be discussed regarding KO is its potential effects on neurological diseases due to the astaxanthin it contains. This is because astaxanthin regulates neuroinflammation by alleviating oxidative stress, reducing the production of neuroinflammatory factors, inhibiting peripheral inflammation, and maintaining the integrity of the blood–brain barrier [63]. Zhang et al. demonstrated that astaxanthin supplementation (25 or 75 mg/kg) in an experimental subarachnoid hemorrhage model can significantly reduce the expression of inflammatory cytokines (reduction in IL-1β and TNF-α expression) and intercellular adhesion molecule-1 expression. Additionally, these changes were noted to lead to improvements in secondary brain injury cascades, including cerebral edema, blood–brain barrier disruption, neurological dysfunction, and neuronal degeneration [64]. Therefore, astaxanthin, when combined with LC n-3, which is naturally found in KO, exhibits antioxidant, anti-inflammatory, and neuroprotective effects in healthy and neurodegenerative research models [61].

### Possible Molecular Mechanism of the Anti-Inflammatory Effects of FO and KO

The molecular mechanisms associated with the anti-inflammatory effects of KO and FO have been attributed to their bioactive components. EPA and DHA are anti-inflammatory FA. The anti-inflammatory effect is achieved by reducing the production of eicosanoid derivatives from arachidonic acid (AA), leukocyte chemotaxic factor formation, adhesion molecule expression, production of pro-inflammatory cytokines, and other pro-inflammatory proteins induced through the NF-κB pathway; inhibiting leukocyte–endothelial adhesion interaction; and increasing the production of anti-inflammatory eicosanoid derivatives from EPA. Changes in the FA composition of the cell membrane mediate the effects of EPA and DHA on inflammatory processes. Changes in FA composition can impact the formation of glycosphingolipid- and cholesterol-rich lipid rafts, cell signaling causing altered gene expression, and the formation of lipid mediators, including eicosanoids and resolvins [65]. During the inflammatory response, cells are rich in AA, an n-6 FA, and EPA and DHA administration can alter its concentrations. EPA and DHA form resolvins, and DHA forms anti-inflammatory protectins and maresins. Increased EPA/DHA and decreased AA content in the membrane lead to a favorable change in eicosanoid, resolvin, and preservative production [65]. A recent study reported that dietary supplementation with 4% DHA or EPA improved hepatic insulin resistance of mice with high-fat-diet-induced insulin resistance by inhibiting hepatic steatosis and activating the PI3K/Akt signaling pathway. DHA and EPA supplementation exerted anti-inflammatory effects by inhibiting TLR4/NF-κB signaling activation [66]. A study on C57BL/6J mice investigated the effects of EPA and AA on inflammation and noted that NF-κB activation was significantly increased by AA but induced by EPA. Moreover, when NF-κB activation was inhibited, MCP-1 secretion became significantly reduced, and inflammatory biomarkers, including IL-6 and angiotensinogen-II, were significantly decreased [67].

The interaction of EPA and DHA with PPAR-γ represents another crucial mechanism by which LC-PUFAs in KO and FO are considered to be associated with inflammation. PPAR-γ, a ligand-dependent transcription factor, is a member of the nuclear receptor superfamily. Serving as sensors of hormones, vitamins, endogenous metabolites, and xenobiotic compounds, nuclear receptors control the expression of multiple genes. Besides its role as a transcription factor, PPAR-γ also functions as a trans-repressor of macrophage inflammatory genes [68]. The ligand-binding domain of PPAR-γ inhibits NF-κB activation by interacting with p65, which is required for NF-κB signaling. PPAR induces increased expression of IκB inhibitory protein-α, sirtuin-1 and phosphatase and tensin homolog (PTEN), which interfere with NF-κB activation and function in inflammatory reactions [69]. The PG synthesis pathway is an essential element of inflammatory responses. At the beginning of the pathway, phospholipase A 2 releases AA from cell membranes. Subsequently, AA is enzymatically converted into a large group of compounds called eicosanoids, initially by lipoxygenases (LOX) or cytochrome P450 (CYP) or best known as COX. Therefore, the synthesis of eicosanoids in mammals occurs from ARA via the COX, LOX, and CYP-450 pathways. The COX enzyme has two forms, including COX-1 and COX-2. The previously described COX-1 enzyme is expressed as a builder of cell membranes in several tissues, whereas the COX-2 enzyme is a trigger of various factors, including cytokines, growth factors, and endotoxins. Increased COX-2 enzyme synthesis is initiated by the activation of NF-κB, known as the universal transcription factor of pro-inflammatory genesis. In acute inflammation, the COX-2 enzyme increases prostaglandin E2 (PGE-2) production in endothelial cells, epithelial cells, stromal cells, monocytes, and lymphocytes to 100 times the basal level. This significant increase results in an exacerbation of the inflammatory response. PPARs are at the center of the COX auto-inhibition pathway. PPARα and PPAR-γ inhibit the NF-κB activity, causing inhibition of inflammatory reactions by the products of these reactions [69,70]. EPA and DHA were noted to bind to PPAR-γ and increase its activation. In a human proximal tubular cell line study, EPA (10 μmol/L) and DHA (100 μmol/L) were observed to effectively reduce LPS-induced NF-kappaB activation and MCP-1 expression. Moreover, EPA and DHA increased PPAR-γ mRNA and protein activity by two–three folds [71]. A study conducted on infarcted rats demonstrated that the administration of EPA and DHA (50 mg/kg) for 4 weeks activated PPAR-γ and reduced the release of IL-1β and neuron growth factor [72]. In a similar study, EPA and DHA supplementation (100 μmol/L) significantly increased PPAR-γ expression and decreased inflammation-related integrin-associated kinase and integrin β1 expression in LPS-treated rat glomerular mesangial cells [73]. A previous meta-analysis investigating the effects of LC-PUFAs in various metabolic diseases evaluated 15 studies and reported that n-3 FA supplementation significantly increased PPAR-γ gene expression compared with the control group [74]. Supplementation with 1000 mg FO (180 mg EPA and 120 mg DHA) twice daily for 6 weeks upregulated PPAR-γ gene expression and reduced IL-1 and TNF-α gene expression in females with gestational diabetes [75]. All these findings suggest that EPA and DHA are PPAR-γ antagonists and can offer insights into the mechanism of the anti-inflammatory effects of FO and KO.

A mechanism explaining the anti-inflammatory effects of LC-PUFAs is G-protein-coupled receptor 120 (GPR120), which serves as an n-3 FA receptor/sensor. A study on mice fed a high-fat diet evaluated the effects of FO (16% EPA and 9% DHA) replacement for 5 weeks and noted that GPR120 acts as an n-3 receptor/sensor and exerts potent and broad anti-inflammatory effects, particularly in macrophages. GPR120 was coupled with β-arrestin2, leading to the inhibition of receptor endocytosis and TAB1 binding protein-mediated activation of growth factor-β-activated kinase 1, ultimately resulting in the inhibition of TLR and TNF-α pro-inflammatory signaling pathways [76].

Unlike FO, KO contains astaxanthin, vitamin E, and vitamin D, contributing to its anti-inflammatory properties [2]. Among these components, astaxanthin has been particularly focused on. Astaxanthin, a precursor of vitamin A, is a lipid-soluble xanthophyll carotenoid noted in various microorganisms and marine animals. According to the literature, astaxanthin possesses antioxidant and anti-inflammatory effects [44]. With these effects, it has been noted to exert protective and therapeutic effects against various diseases, including neurological diseases, diabetes, gastrointestinal diseases, liver and kidney diseases, eye conditions, and skin disorders. Moreover, astaxanthin has anti-inflammatory effects. It suppresses inflammation by reducing pro-inflammatory cytokine production and inhibiting the NF-κB and mitogen-activated protein kinase pathways [44]. Furthermore, astaxanthin inhibited nitrite oxide (NO), PGE-2, and TNF-α production and decreased inducible NO synthase (iNOS) activity in LPS-stimulated cells [77]. In an in vivo study by Santos et al., astaxanthin administration inhibited O_2_ •-, NO, and TNF-α production in rat alveolar macrophages [78].

## 3. Comparison of Antioxidant Effects of FO and KO

Studies investigating KO and oxidative stress have reported remarkable findings. It was noted that 40 mg/kg KO supplementation for 8 weeks reduced oxidative stress by improving thiobarbituric acid reactive substances (TBARS) and glutathione (GSH) levels in the liver and spleen of iron-loaded rats [79]. In experimental nephrotoxicity model rats, 500 mg/kg KO supplementation for 7 days improved plasma malondialdehyde (MDA) levels and caused histopathologic improvements [80]. A study in rowing athletes demonstrated that 1 g/day of KO supplementation accelerated post-exercise recovery and reduced oxidative stress [81]. A study conducted on a mouse model of multiple sclerosis showed that feeding a KO-rich diet for 5 weeks increased GSH peroxidase, superoxide dismutase (SOD) activities, and GSH levels; decreased MDA concentrations; and suppressed inflammation [82]. In a study of the effects of 2 g/day KO supplementation for 6 weeks on exercise performance and immune system in healthy adults, KO supplementation did not affect plasma TBARS, IL-6, IL-7, and interferon-γ levels [83]. In a placebo-controlled double-blind randomized trial in athletes, supplementation with 2.5 g KO (550 mg EPA/DHA and 150 mg choline) for 12 weeks significantly increased the high-sensitive n-3 index and antioxidant capacity and decreased the ARA/EPA ratio [84]. In a study conducted on an LPS-induced Alzheimer’s disease model, KO-rich diet (80 mg/kg/day) fed to mice for 1 month inhibited iNOS and COX-2 expression and decreased tissue ROS and MDA levels [85]. In a study investigating the effects of KO on gastric ulcers, rats were administered 100, 200, or 500 mg/kg of KO by oral gavage for 1 month before experimental ulcer formation. The results of the experiment showed that KO administration at 500 mg/kg inhibited ulcer formation and decreased tissue MDA levels and myeloperoxidase (MPO) activity. Moreover, SOD activity increased and TNF-α, IL-6, and IL-1β expressions decreased with KO administration [86].

In the literature, increasing evidence is available on the association between FO and oxidative stress. In a meta-analysis investigating the effects of n-3 supplementation on oxidative stress parameters, total antioxidant capacity (TAC), and GPx activity significantly increased, while serum MDA levels decreased with n-3 supplementation. In this meta-analysis, n-3 supplementation did not affect NO, GSH, SOD, and catalase (CAT) levels [87]. In a streptozotocin-induced diabetes model, 4-μg/g FO supplementation via the orogastric route for 8 weeks decreased cardiac MPO activity and increased GSH, CAT, and SOD activities [88]. Shaaban et al. noted that oral n-3 supplementation at different doses (75 and 150 mg/kg) for 12 weeks caused improvement in MDA, GSH levels, and MPO activity and suppressed oxidative stress in rats with hepatic fibrosis [89]. In acute cold restraint stress-induced liver injury model rats, 0.4 g/kg of n-3 supplementation via oral gavage for 4 weeks significantly decreased hepatic MDA levels. Furthermore, increased TAC levels and decreased serum corticosterone, alanine aminotransferase, aspartate aminotransferase (AST), and TNF-α levels were observed [90]. A double-blind randomized controlled trial conducted on patients with Alzheimer’s disease revealed that FO supplementation (0.45 g EPA and 1 g DHA) for 12 months reduced plasma levels of lipoperoxide and nitric oxide catabolites, which was associated with a higher GSH/GSSG ratio in plasma and reduced oxidative stress [91]. In a double-blind randomized controlled trial conducted on healthy older adults, participants received FO (6 g/day), FO (6 g/day) + multivitamin, FO (3 g/day) + multivitamin, or placebo for 16 weeks. No significant change in inflammatory biomarkers, including IL-6, IL-Iβ, and TNFα, was noted following administration, whereas the levels of plasma F2-isoprostane, an oxidative stress marker, significantly decreased [92].

Studies comparing the effects of KO and FO on oxidative stress have demonstrated consistent results. Some studies examining the antioxidant and anti-inflammatory effects of KO and FO in the literature are summarized in Table 1. Polotow et al. investigated the effects of omega-3 PUFAs and astaxanthin supplementation on redox status and neuroinflammation in brain tissue. Rats were supplemented with 10 mg/kg EPA + 7 mg/kg DHA or 1 mg/kg astaxanthin or FO + astaxanthin (10 mg/kg EPA + 7 mg/kg DHA and 1 mg/kg astaxanthin) or KO (10 mg/kg EPA + 4.7 mg/kg DHA + 0.0072 mg/kg astaxanthin) via oral gavage for 45 days. Although no difference in TBARS level, GPx, glutathione reductase (GR), CAT, and manganese superoxide dismutase (MnSOD) activity in cerebellar tissues was noted between the groups, astaxanthin supplementation decreased the enzyme activity. Compared with the control group, KO supplementation did not affect GSH, GSSG, TNF-α, IL-6, and IL-1β levels [93]. A study on healthy volunteers showed that supplementation with FO (3 g/day, EPA + DHA = 543 mg) or KO (1.8 g/day, EPA + DHA = 864 mg) for 7 weeks made no difference in terms of oxidative stress biomarkers (α-tocopherol and urinary F2-isoprostane). Null effects on urinary F2-isoprostanes may reflect inadequate dosing or short duration for biomarker turnover [13]. In an experimental study in rats, the effects of feeding a diet rich in various n-3 fats (corn oil, flaxseed oil, KO, menhaden oil, salmon oil, or tuna oil) with concentrations of 7–12% for 8 weeks on metabolic and oxidative stability was investigated. The KO-fed group exhibited lower DHA digestibility and brain DHA incorporation than rats fed salmon and tuna oil-rich diets, whereas tissue accumulation was higher in the salmon and tuna group. Despite this finding, no significant increase in TBARS or increased susceptibility to oxidation as indicated by a decrease in TAC and gene expression of antioxidant defense enzymes was observed in the groups [54]. In a study conducted on rats with a stress model, they received 0.5 g/kg Neptune KO, 0.32 g/kg FO, or 1 mg/kg B12 supplementation once a day for 14 days. KO and FO reduced MDA, H_2_O_2_ levels at similar levels, whereas GSH, SOD, GPx, and CAT antioxidant enzyme levels were regulated better than B12 without significant difference between them [94]. In a study conducted on an ethanol-induced experimental ulcer model, 2.5% KO and FO supplementation for 4 weeks showed a similar protective effect against ulcer damage as it inhibited ROS formation and decreased lipid peroxidation [55].

### Possible Molecular Mechanism of the Antioxidant Effects of FO and KO

The primary focus on the antioxidant mechanism of KO and FO is the effects of the n-3 PUFAs they contain. As previously mentioned, in KO, EPA, and DHA are in the form of phospholipids (>80% phosphatidylcholine), whereas in FO, they are in the form of triglycerides. Phosphatidylcholine has effects on homocysteine levels, improving liver diseases and reducing respiratory distress [5,95]. However, in the literature, the impact of the difference between the FA forms contained in KO and FO has primarily been evaluated in terms of bioavailability and has not been emphasized in terms of oxidative stress or inflammation [5,95]. Considering the possible beneficial effects of FA on oxidative stress, this is a gap that requires clarification.

KO is believed to exert its antioxidant effect through various pathways. In a placebo-controlled study on individuals with coronary heart disease, participants were supplemented with 2 g/day of KO for 3 months, and its effects were analyzed. KO supplementation increased antioxidant capacity by decreasing oxidative biomarkers ROS, 8-hydroxy-2′-deoxyguanosine, NO, and MDA levels and increasing antioxidant biomarkers SOD, GSH, and GPx enzyme activities. Evaluating all findings revealed that EPA and DHA exerted this effect by activating KEAP1 and NRF2 signaling. Although KEAP1 and NRF2 are associated with antioxidant function, KEAP1/NRF2 signaling activation causes increased antioxidant activities [96]. In a study evaluating the effects of KO on methamphetamine-induced neurotoxicity, PC12 cells were exposed to both methamphetamine (3 mmol/L) and KO (0.1, 0.2, 0.4, and 0.8 μg/mL) for 24 h in vitro. KO decreased oxidative stress by increasing SOD and GSH activity and decreasing MDA, NO, and ROS generation. Although the mitochondrial mechanism has not been fully elucidated, KO demonstrated a beneficial effect on the apoptotic response by increasing cell viability, decreasing protein expression of cleaved caspase-3, and decreasing mitochondrial membrane potential dissipation [97]. A study on aged Nothobranchius guentheri fish showed that KO-rich diet (40 mg/day) decreased protein oxidation, lipid peroxidation, and ROS levels as well as increased SOD, CAT, and GPx enzyme activities in the liver. Moreover, KO suppressed the NF-κB signaling pathway, the main regulator of inflammation [98].

As previously reported, KO contains astaxanthin. Evidence from literature underscores the antioxidant and anti-inflammatory effects of astaxanthin. With these effects, astaxanthin has been observed to exert protective and therapeutic effects against several diseases, including neurological diseases, diabetes, gastrointestinal diseases, liver and kidney diseases, eye conditions, and skin disorders. Astaxanthin, a xanthophyll carotenoid, has a molecular structure containing hydroxyl and keto moieties with high antioxidant properties on each ionone ring [44,99]. Conjugated double bonds in carotenoids (e.g., astaxanthin) dissipate singlet oxygen energy as heat, terminating oxidation cascades. Astaxanthin’s polyene chain quenches lipid peroxyl radicals within membranes, while its polar end groups scavenge aqueous radicals at membrane interfaces [44,99]. In another study conducted with naxopen in an experimental ulcer model, 25 mg/kg astaxanthin supplementation generated a protective effect against peptic ulcer, inhibited the increase in lipid peroxidation products in the gastric mucosa, and increased the activities of essential antioxidant enzymes (SOD, CAT, and GPx) [100]. The beneficial effects of astaxanthin on gastric ulcers are attributed to ROS inhibition and antimicrobial activity by inhibiting bacterial growth during *Helicobacter pylori* (*H. pylori*) infection. Furthermore, astaxanthin precludes immune function impairment by shifting T cell 1 (Th1) response to Th2 response, thereby suppressing gastric inflammation [101]. In patients with *H. pylori*, an 8-week 40 mg astaxanthin supplementation did not alter inflammatory cytokine levels, but it enhanced humoral immunity (through CD4 expression upregulation and CD8 expression downregulation) [102]. Other significant mechanisms proposed for the antioxidant effects of astaxanthin are through the NF-κB and NRF2 pathways. NRF2 modulates antioxidant response elements in the promoter regions of AST, the most cytoprotective or detoxifying enzyme. NRF2 induces antioxidant enzymes (HO-1, NQO1), it is also a key regulator of intracellular redox homeostasis maintenance. By modulating the NF-κB signaling network, astaxanthin ameliorates inflammation and oxidative stress [103]. A previous study reported that astaxanthin treatment increased the NRF2, heme-oxygenase 1, and MnSOD expression, while significantly decreasing the Keap1 and NF-κB expression, indicating antioxidant and anti-inflammatory effects [104]. The antioxidant and anti-inflammatory mechanisms of KO and FO are summarized in Figure 2.

## 4. Materials and Methods

All relevant studies up to 2025 were identified through a systematic search using key search terms in the Science Direct, PubMed, and Web of Science databases. Identified search terms include “fish oil,” “krill oil,” “oxidative stress,” “inflammation,” “antioxidant,” and “anti-inflammatory.”

The inclusion criteria for studies were as follows: (1) in vitro, in vivo, human, or animal studies; (2) participants receiving krill and/or fish oil supplements; (3) participants receiving a krill and/or fish oil diet; and (4) examining biomarkers related to oxidative stress and inflammation. Exclusion criteria were the following: (1) studies that did not examine oxidative stress and inflammatory biomarkers; (2) studies that remained incomplete, lacked full-text articles, or had withdrawn registrations. Articles with full text, meeting the search criteria and in English were included in the study.

## 5. Conclusions

This review elucidates how compositional differences (e.g., phospholipids vs. triglycerides, astaxanthin vs. tocopherols) dictate distinct molecular actions. Scientific studies have revealed that KO and FO show antioxidant and anti-inflammatory effects in in vitro studies as well as in vivo human and experimental animal models. Although KO and FO exhibit biological and physiological effects through their bioactive components, they occasionally exert similar effects and sometimes show superiority over each other. In this review, the molecular mechanisms of the antioxidant and anti-inflammatory effects of FO and KO are also elucidated. Deciphering KO/FO pathway-specific efficacy will advance precision nutraceutical applications for inflammatory and oxidative pathologies. Future studies should quantify dose- and tissue-dependent differences in KO/FO pathway modulation (e.g., NRF2 activation kinetics, resolvin synthesis) to optimize clinical applications.

## Figures and Tables

**Figure 1 ijms-26-07360-f001:**
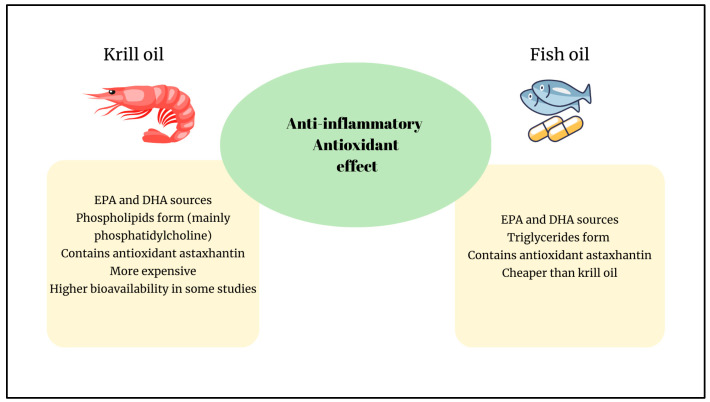
Main differences between KO and FO. EPA: eEicosapentaenoic acid; DHA: docosahexaenoic acid.

**Figure 2 ijms-26-07360-f002:**
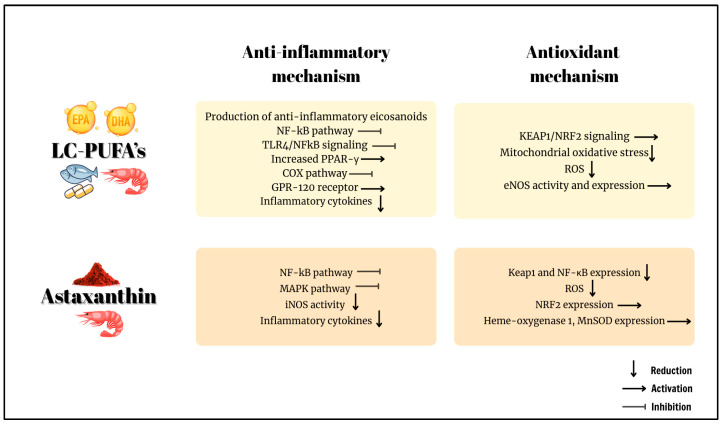
Antioxidant and anti-inflammatory mechanisms of KO and FO. NF-κB: nuclear factor kappa B; TLR4: toll-like receptor-4; PPAR-γ: PPAR-gamma; COX: cyclooxygenase; GPR-120: G-protein-coupled receptor 120; KEAP1/NFR2: Kelch-like ECH-associated protein 1/nuclear factor erythroid 2-related factor-2; ROS: reactive oxygen species; eNOS: endothelial nitric oxide synthase; MAPK: mitogen-activated protein kinase; iNOS: inducible nitric oxide synthase; MnSOD: manganese superoxide dismutase.

**Table 1 ijms-26-07360-t001:** Some studies comparing the anti-inflammatory and antioxidant effects of krill oil and fish oil.

Author	Study Design	Sample	Intervention	Amount of n-3 PUFAs and Others	Results
Ierna et al., 2010 [49]	Experimental study	DBA/1 mice modeled for arthritis *n* = 42	68 days of supplementation.	0.44 g/100 g KO or FO	More decrease in swelling in arthritis and arthritis disease score in the KO group. No change in serum cytokines in the KO group. Higher levels of serum IL-1α and IL-13 in the FO group.
Batetta et al., 2009 [51]	Experimental study	Obese male zucker rats *n* = 18	Diet for 4 weeks.	KO or FO-rich diet containing 0.5 g EPA + DHA/100 g	Similar decrease in LPS-induced TNF-α release from peritoneal macrophages. Improvement in TAG concentrations in the KO group.
Vigerust et al., 2013 [53]	Experimental study	Transgenic C57BL/6 hTNF-α mice *n* = 26	Diet for 6 weeks.	Control diet with 24.50% total fat or high-fat diets with FO or KO	Reduced inflammation in both groups, although KO was more effective. Increase in IL-17 levels in mice in FO group. Decrease in MCP-1 levels in KO group. Pro-inflammatory cytokine levels (IL)-1b, IL-2, IL-17, and IFN-γ were not significantly different between treatment groups.
Cicero et al., 2016 [52]	Double-blind, crossover, randomized clinical trial.	Moderately hypertriglyceridemic individuals *n* = 25	4 weeks of supplementation.	1000 mg omega-3 ethyl ester PUFA or 500 mg KO twice daily	Significant improvement in high density lipoprotein cholesterol and apolipoprotein AI levels in the KO group. Decrease in hs-CRP in the KO group.
Sarıyer et al., 2024 [55]	Experimental study	Sprague-Dawley rats subjected to experimental ulcer induced *n* = 64	4 weeks of supplementation.	KO, FO or ASX at 2.5% (*v*/*w*)	Similar results in biomarkers of inflammation and oxidative stress in KO and FO groups. KO, FO, and ASX supplementation reduced chemiluminescence levels in the ulcer group, while only ASX supplementation decreased MDA levels and MPO activity. Better results in ROS inhibition and lipid peroxidation prevention in the ASX group.
Zadeh-Ardabili et al., 2019 [50]	Experimental study	Adult male Swiss mice Carrageenan induced inflammation in mice models *n* = 168	Once	500 mg NKO or FO (balanced at similar doses of EPA: 12 in NKO vs. 12 in FO wt%, DHA: 7 NKO vs. 8 FO wt%)	In the NKO group, 43.6% inhibition of edema at 3 h. In FO group, 35.1% inhibition at the same time In both groups TNF-α and IL-6 levels were significantly decreased, but the effect of NKO was more pronounced.
Polotow et al., 2015 [93]	Experimental study	Wistar rats *n* = 40	45 days of supplementation.	10 mg EPA/kg + 7 mg DHA/kg FO or 1 mg/kg ASX or 10 mg EPA/kg + 7 mg DHA/kg + 1 mg ASTA/kg (FO + ASX) or 10 mg EPA/kg + 4.7 mg DHA/kg + 7.2 μg ASTA/kg (KO + ASX)	No difference in cerebellum tissue TBARS level, TEAC, GPx, GR, or CAT activity between the groups. Mitochondrial MnSOD activity decreased by 50% and 59.6% in ASX and FO + ASTA groups compared to the control group, respectively. No change in GSH, GSSG, TNF-α, IL-6, or IL-1β levels in the KO group compared to the control group. FRAP was 2.1 times higher in the ASX group compared to the control group.
Ulven et al., 2011 [13]	Open-label, single-center, randomized, parallel-group study	Individuals with normal or mildly elevated total blood cholesterol and/or triglyceride levels *n* = 113	7 weeks of supplementation.	FO (3 g/day, EPA + DHA = 543 mg) or KO (1.8 g/day, EPA + DHA = 864 mg)	No difference in MDA or TAC levels. No difference in IL-6, TNF-a, or CRP levels. Significant increase in plasma EPA, DHA, and DPA levels were similar between groups.
Zadeh-Ardabili et al., 2019 [94]	Experimental pilot study	Adult male Swiss albino mice. Stress model induced using natural light. *n* = 108	14 days of supplementation.	Once a day; 0.5 g/kg Neptün KO, 0.32 g/kg FO or 1 mg/kg B12 or 10 mg/kg/day imipramin	Similar decrease in MDA, H_2_O_2_ levels between KO and FO groups. No significant difference in GSH, SOD, GPx and CAT antioxidant enzyme levels between KO and FO groups, but more effective results than B12.

KO: krill oil; FO: fish oil; IL: interleukin; EPA: eicosapentaenoic acid; DHA: docosahexaenoic acid; LPS: lipopolysaccharide; TNF-α: tumor necrosis factor-alpha; TAG: triacylglycerol; MCP-1: monocyte chemotactic protein-1; IFN-γ: interferon-gamma; PUFA: polyunsaturated fatty acids; hs-CRP: high-sensitive CRP; ASX: astaxanthin; MPO: myeloperoxidase; MDA: malondialdehyde; NKO: Neptün krill oil; TBARS: thiobarbituric acid reactive substances; TEAC: trolox-equivalent antioxidant capacity; GPx: glutathione peroxidase; GR: glutathione reductase; CAT: catalase; FRAP: iron reducing capacity; TAC: total antioxidant capacity; DPA: docosapentaenoic acid; H_2_O_2_: hydrogen peroxide.

## Data Availability

Not applicable.

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
