# Peer review of "Comparative Analysis of the Antioxidant and Anti-Inflammatory Effects of Krill and Fish Oil"

_ijms, 2025, doi:10.3390/ijms26157360_

Round 1

Reviewer 1 Report

Comments and Suggestions for Authors

Although a review on only two molecules may seem reductive it is still interesting for the topic itself. In fact EFSA only in 2023 gave its opinion on the consumption of Krill oil since it is classified as novel food. The comparison among properties of Krill and Fish oils are interesting for the lectors and I suggest to accept the manuscript. However some minor changes could improve the quality of manuscript: I propose to reduce some of the information in paragraphs 2 and 3 and introduce a new paragraph on what is known about metabolites produced in humans and the pharmacokinetics of the two oils.

  • For example delete from 76 to 82 and from 102 to 109.
  • From 280 to 292 and from 379 to 395.

Information of this type is now sufficiently known, while it can be very useful to have information on human pharmacokinetics.

Author Response

Dear Reviewer 1,

Thank you for your suggestions and contributions. The explanation of the corrections you requested is provided below and is highlighted in blue in the text file.

Comment 1. I propose to reduce some of the information in paragraphs 2 and 3 and introduce a new paragraph on what is known about metabolites produced in humans and the pharmacokinetics of the two oils.

  • For example delete from 76 to 82 and from 102 to 109.
  • From 280 to 292 and from 379 to 395.

Response 1

  • Thank you for pointing this out. We agree with this comment. We have deleted the places you suggested from the article.

Comment 2. Introduce a new paragraph on what is known about metabolites produced in humans and the pharmacokinetics of the two oils.

  • Response 2. Thank you for pointing this out. We agree with this comment. We added new paragraphs in the article.

Reviewer 2 Report

Comments and Suggestions for Authors

Please check the attached document.

Author Response

Dear Reviewer 2,

Thank you for your suggestions and contributions. The explanation of the corrections you requested is provided below and is highlighted in yellow in the text file.

Comment 1: Please add a new section  “2. Methodology” to strengthen the structure and transparency of the review. This section could outline the criteria used for article selection (e.g., databases searched, keywords, inclusion/exclusion criteria, and timeframe). Additionally, I suggest including a graphical representation (such as a timeline or bar chart) showing the number of articles published in recent years on the antioxidant and anti-inflammatory effects of both krill and fish oil. This would provide readers with a clear overview of the research trends in the field.

Response 1: Thank you for pointing this out. We agree with this comment. We added the Methodology section. Since our article is not a systematic review, we did not add the article numbers or a diagram to avoid confusion.

Comment 2: I recommend adding a new section titled “3. Extraction of Bioactive Compounds from Krill and Fish Oil.” and a new Table. This section should describe the primary extraction methods and conditions used to recover key compounds such as LC-PUFAs (EPA, DHA), and astaxanthin. Please include relevant parameters such as temperature, extraction time, type of solvents or technologies used (e.g., supercritical fluid extraction, solvent extraction), and any purification steps. Including this information will enhance the technical depth of the review and provide valuable insight into the efficiency and selectivity of current extraction approaches.

Response 2. : Thank you for pointing this out. We agree with your comment. We added this section after talking about bioavailability in the introduction. However, we did not add a table so as not to distract from the main topic. We believe the general information provided is sufficient.

Comment 3. Lines 89 and 95. Kindly italicize 'in vitro' for clarity.

Response 3: Thank you for your attention. We have made corrections to the entire text. Line numbers may have changed due to additions and deletions in the text.

Reviewer 3 Report

Comments and Suggestions for Authors

This work turns relevant for being more comprehensive comparing to the last reviews assessed recently in 2024 about the potential of KO are oriented for knee pain and bowel inflammation, whereas the specific comparison between the effect of KO and FO is regarding the specific cardiovascular effect (DOI:10.1016/j.jff.2024.106379 ).

62% of references are over 5 years ago. Althout the autonomy of citation belongs to the authors, I kindly invite to, if possible, add more recent references to the already mentioned, to give this review a sound and updated state of art on the research question. Please, also make sure that all the references are in accord with the Journals requirements.

Regarding the Abstract, it is succint and well written however i would like to invite the authors to include a brief description of the methodology (once sentence) applied to achieve their results in this review. I also suggest adding a graphical abstract to appeal to the attention of the readers potentially interested in this field.

Figure 1 has a good resolution I just kindly suggest centering it in the page.

Lines 32-33: "on various health conditions, particularly cardiovascular diseases. " I kindly invite the authors to discriminate more health conditions that can be improved with these derivatives.

Lines 47-48: " α-Linolenic acid (ALA), EPA and DHA represent the main n-3 fatty acids" - Please add the respective recent reference for supporting these data.

Line 68 and 69: Is it possible to add a more recent reference, besides this one [8 and 10]? 2011 is almost 15 years ago...

Line 74: I strongly suggest adding a resume of the methodology used to structure and find these results. For eg. a boylean stringline for bibliographic search, and how the articles were included or discarded according to the abstract/full text reading. How many researchers involved in the bibliographic research. Types of databases used; timeline considered. Languages included for article selection (If only English or more), etc. Something along PRISMA, for eg. (although it is clearly a non systematic review, it would help to follow the line of scientific through of the authors and imprint more credibility to the study).

Line 82: Please, include a recent reference to support this sentence.

Lines 89, 95 and throughout the text: Please format "in vitro" to italics.

Line 111 and throughout the text: Please format "et al." to italics and add a comma plus year of publication.

Lines 150-153: Is it possible to mention the reasons the authors of that study alledge for these differences?

Line 167-169: Is it possible to discriminate the profile of lipids that were lowered with the treatment?

In this point 2 of the manuscript I suggesting adding more information about the potential to reduce neuroinflammation by these oils in the neurodegenerative diseases such as Alzheimer and Parkinson's, and neurodevelopmental conditions with some grade inflammation such as within autism spectrum. To simplify reading I would suggest contracting fatty acid to FA.

Lines 282-285: "all lead to endogenous ROS formation." Please add a reference to support this information.

Table 1: I suggest choosing a timeframe and searching all the studies comparing the anti-inflammatory and antioxidant effects of KO and FO, to include in the table.

I strongly motivate the authors to introduce a picture elucidating the main limitation of this study and their study suggestions for discriminated main study directions.

Overall the study is very well written and, according to my perspective, the English is quite good.

Author Response

Dear Reviewer 3,

Thank you for your suggestions and contributions. The explanation of the corrections you requested is provided below and is highlighted in red in the text file.

Comment 1. Regarding the Abstract, it is succinct and well written however I would like to invite the authors to include a brief description of the methodology (once sentence) applied to achieve their results in this review.

Response 1. Thank you for pointing this out. We agree with this comment. We added the Methodology section. You can see it as the 4th heading in the text.

Comment 2. Figure 1 has a good resolution I just kindly suggest centering it in the page.

Response 1. Agree. We have changed.

Comment 3. Lines 32-33: "on various health conditions, particularly cardiovascular diseases. " I kindly invite the authors to discriminate more health conditions that can be improved with these derivatives.

Response 3. Thank you for pointing this out. We agree with this comment. We have revised.

Comment 4. Lines 47-48: " α-Linolenic acid (ALA), EPA and DHA represent the main n-3 fatty acids" - Please add the respective recent reference for supporting these data.

Response 4. Thank you for pointing this out. We added recent reference.

Comment 5. Line 68 and 69: Is it possible to add a more recent reference, besides this one [8 and 10]?

Response 5. Thank you for pointing this out. We added recent reference.

Comment 6. Line 74: I strongly suggest adding a resume of the methodology used to structure and find these results. For eg. a boylean stringline for bibliographic search, and how the articles were included or discarded according to the abstract/full text reading. How many researchers involved in the bibliographic research. Types of databases used; timeline considered. Languages included for article selection (If only English or more), etc. Something along PRISMA, for eg. (although it is clearly a non-systematic review, it would help to follow the line of scientific through of the authors and imprint more credibility to the study).

Response 6. Thank you for pointing this out. We agree with this comment. We have revised.

Comment 7. Line 82: Please, include a recent reference to support this sentence.

Response 7. This section was removed from the text on the suggestion of another reviewer.

Comment 8. Lines 89, 95 and throughout the text: Please format "in vitro" to italics.

Response 8. Thank you for your attention. Line numbers may have changed due to additions and deletions in the text. We have made corrections to the entire text.

Comment 9. Line 111 and throughout the text: Please format "et al." to italics and add a comma plus year of publication.

Response 9. Thank you for your attention. We have revised.

Comment 10. Lines 150-153: Is it possible to mention the reasons the authors of that study alleged for these differences?

Response 10. Thank you for your suggestion. We have added the clarification to the text.

Comment 11. Line 167-169: Is it possible to discriminate the profile of lipids that were lowered with the treatment?

Response 11. Thank you for your attention. We have revised.

Comment 12. In this point 2 of the manuscript I suggesting adding more information about the potential to reduce neuroinflammation by these oils in the neurodegenerative diseases such as Alzheimer and Parkinson's, and neurodevelopmental conditions with some grade inflammation such as within autism spectrum. To simplify reading I would suggest contracting fatty acid to FA.

Response 12. Thank you for your attention. Suggestions have been made and added to the text. The term fatty acid is abbreviated as FA throughout the text. We have also included a summary of the literature on neurological diseases and neuroinflammation (Line 257-301).

Comment 13. Lines 282-285: "all lead to endogenous ROS formation." Please add a reference to support this information.

Response 13. This section was removed from the text on the suggestion of another reviewer.

Comment 14. I suggest choosing a timeframe and searching all the studies comparing the anti-inflammatory and antioxidant effects of KO and FO, to include in the table.

I strongly motivate the authors to introduce a picture elucidating the main limitation of this study and their study suggestions for discriminated main study directions.

Response 14. Thank you for this excellent and constructive suggestion. Our primary purpose in writing this review stemmed from the scientific studies conducted using various KO and FO supplements at different doses, for different durations, and in different models. The most effective and/or recommended compound remains unconfirmed. With this review, we aimed to highlight these differences. Figure 2 was prepared based on information obtained from the publications used in this review.

Reviewer 4 Report

Comments and Suggestions for Authors

Manuscript ID: ijms-3760211

Manuscript Title: Comparative Analysis of the Antioxidant and Anti-inflammatory Effects of Krill and Fish Oil

Journal Name: International Journal of Molecular Sciences

Reviewer Comments

Subject and title

The topic of the manuscript is relevant and interesting for the journal audience.

Abstract

Line 16: The opening sentence implies structural differences cause health effects rather than being delivery mechanisms. Rephrase to: ‘Krill oil (KO) and fish oil (FO) are rich sources of long-chain polyunsaturated fatty acids, with eicosapentaenoic acid (EPA) and docosahexaenoic acid (DHA) bound to distinct molecular carriers (phospholipids vs. triglycerides)’

Line 18: The phrase ‘roles in health maintenance and potential therapeutic benefits’ is redundant with the title’s focus. Replace with: ‘...extensively studied for their antioxidant and anti-inflammatory properties relevant to disease prevention/therapy’

Line 22: (Both oils demonstrate biological and physiological activities) is unspecific. Correct to : ‘Both oils exert antioxidant and anti-inflammatory activities’ to aligns with review focus.

Line 23: I suggest adding this statement ‘The bioactivities of both oils stem from their distinct molecular compositions: KO delivers EPA/DHA via phospholipids alongside astaxanthin, while FO provides triglyceride-bound EPA/DHA’

Line 25-26: The conclusion lacks specificity about research gaps. Expand to: e.g., ‘Further comparative studies examining dose-dependent effects, bioavailability kinetics, and tissue-specific molecular pathways are warranted.’

1. Introduction

I suggest adding chemical structure for the main components such as: EPA, DHA, and astaxanthin

Line 31-32: Add recent citations here

Line 37, 46, 56, 57: Add references

Line 41: (KO contains 30% astaxanthin) is inaccurate. Please correct. You can see articles:

https://doi.org/10.1016/j.lwt.2024.116429

https://doi.org/10.1111/1541-4337.12427

https://doi.org/10.1007/s11814-011-0186-2

line 71: No clear hypothesis/knowledge gap stated. Add: e.g., ‘Despite compositional differences, systematic comparisons of their mechanistic actions on inflammatory/redox pathways remain limited.’

2. Comparison of Anti-inflammatory Effects of FO and KO

Line 125-127: Rewrite: ‘Hence, KO’s anti-inflammatory activity arises synergistically from: (i) phospholipid-bound EPA/DHA modulating eicosanoid pathways, (ii) astaxanthin inhibiting NF-κB/MAPK, and (iii) vitamin E stabilizing membranes [23,48]’

Line 150-152: (KO reduced arthritis scores more than FO) without quantifying. Consider: ‘KO decreased arthritis scores by …% vs. FO’s ….% in mice [28].’

Line 174-176: (FO increased IL-17 [32]) without mechanistic context, e.g., correct to ‘FO-induced IL-17 elevation may reflect n-3 PUFA modulation of Th17 differentiation [32].’

2.1. Possible Molecular Mechanism of the Anti-inflammatory Effects of FO and KO

Line 190-200: Add references

Line 267-268: Add reference

Line 276: Words like ‘in vitro’ and ‘in vivo’ are italicized and unitalicized in manuscript. One format must be used.

3. Comparison of Antioxidant Effects of FO and KO

Line 280-283: Add references

Line 354: Human studies (e.g., Ulven et al., 2010) reported without mechanistic context. Suggestion adds, e.g., : ‘Null effects on urinary F2-isoprostanes [68] may reflect inadequate dosing or short duration for biomarker turnover.’

Line 369: In Table 1, there is no need to mention the year of publication with the authors’ names. The reference number is sufficient.

3.1. Possible Molecular Mechanism of the Antioxidant Effects of FO and KO

Line 427: Add references

Line 429-430: Rephrase to: ‘Conjugated double bonds in carotenoids (e.g., astaxanthin) dissipate singlet oxygen energy as heat, terminating oxidation cascades [83].’

Line 430-432: Rephrase to: ‘Astaxanthin’s polyene chain quenches lipid peroxyl radicals within membranes, while its polar end groups scavenge aqueous radicals at membrane interfaces [83, 84]’

Line 444-445: Please revise (NRF2 regulates AST), ‘NRF2 induces antioxidant enzymes (HO-1, NQO1)’

4. Conclusion

Line 460: The conclusion does not revisit the molecular focus promised in the Introduction. Reframe opening: ‘This review elucidates how compositional differences (e.g., phospholipids vs. triglycerides, astaxanthin vs. tocopherols) dictate distinct molecular actions.’

Line 463, 468: Tables and figures should not be referenced in the Conclusion section. Relevant sentences should be rephrased.

Line 468: I suggest adding: ‘Deciphering KO/FO pathway-specific efficacy will advance precision nutraceutical applications for inflammatory and oxidative pathologies. Future studies should quantify dose- and tissue-dependent differences in KO/FO pathway modulation (e.g., NRF2 activation kinetics, resolvin synthesis) to optimize clinical applications.’

Author Response

Dear Reviewer 4,

Thank you for your suggestions and contributions. The explanation of the corrections you requested is provided below and is highlighted in pink in the text file.

Comment 1. Line 16: The opening sentence implies structural differences cause health effects rather than being delivery mechanisms. Rephrase to: ‘Krill oil (KO) and fish oil (FO) are rich sources of long-chain polyunsaturated fatty acids, with eicosapentaenoic acid (EPA) and docosahexaenoic acid (DHA) bound to distinct molecular carriers (phospholipids vs. triglycerides)’

Response 1. Thank you for pointing this out.We corrected it in the text.

Comment 2.  Line 18: The phrase ‘roles in health maintenance and potential therapeutic benefits’ is redundant with the title’s focus. Replace with: ‘...extensively studied for their antioxidant and anti-inflammatory properties relevant to disease prevention/therapy’.

Response 2. Thank you for pointing this out.We corrected it in the text.

Comment 3. Line 22: (Both oils demonstrate biological and physiological activities) is unspecific. Correct to : ‘Both oils exert antioxidant and anti-inflammatory activities’ to aligns with review focus.

Response 3. Thank you for your suggestion. We corrected it in the text.

Comment 4. Line 23: I suggest adding this statement ‘The bioactivities of both oils stem from their distinct molecular compositions: KO delivers EPA/DHA via phospholipids alongside astaxanthin, while FO provides triglyceride-bound EPA/DHA’.

Response 4. Thank you for your suggestion. We added it in the text.

Comment 5. Line 25-26: The conclusion lacks specificity about research gaps. Expand to: e.g., ‘Further comparative studies examining dose-dependent effects, bioavailability kinetics, and tissue-specific molecular pathways are warranted.’

Response 5. Thank you for pointing this out.We corrected it in the text.

Comment 6. I suggest adding chemical structure for the main components such as: EPA, DHA, and astaxanthin.

Response 6. Thank you for your suggestion. We added it in the text.

Comment 7. Line 31-32: Add recent citations here. Line 37, 46, 56, 57: Add references.

Response 7. Thank you for your suggestion. We added it in the text.

Comment 8. Line 41: (KO contains 30% astaxanthin) is inaccurate. Please correct. You can see articles:

https://doi.org/10.1016/j.lwt.2024.116429

https://doi.org/10.1111/1541-4337.12427.

https://doi.org/10.1007/s11814-011-0186-2

Response 8. Thank you for pointing this out.We corrected it in the text.

Comment 9. line 71: No clear hypothesis/knowledge gap stated. Add: e.g., ‘Despite compositional differences, systematic comparisons of their mechanistic actions on inflammatory/redox pathways remain limited.’

Response 9. Thank you for pointing this out.We revised it in the text.

Comment 10. Line 125-127: Rewrite: ‘Hence, KO’s anti-inflammatory activity arises synergistically from: (i) phospholipid-bound EPA/DHA modulating eicosanoid pathways, (ii) astaxanthin inhibiting NF-κB/MAPK, and (iii) vitamin E stabilizing membranes [23,48]’

Line 150-152: (KO reduced arthritis scores more than FO) without quantifying. Consider: ‘KO decreased arthritis scores by …% vs. FO’s ….% in mice [28].’

Line 174-176: (FO increased IL-17 [32]) without mechanistic context, e.g., correct to ‘FO-induced IL-17 elevation may reflect n-3 PUFA modulation of Th17 differentiation [32].’

Response 10. Thank you for pointing this out.We revised all in the text.

Comment 11. Line 190-200: Add references

Line 267-268: Add reference

Line 276: Words like ‘in vitro’ and ‘in vivo’ are italicized and unitalicized in manuscript. One format must be used.

Response 11. Thank you for pointing this out.We corrected all  in the text.

Comment 12. Line 280-283: Add references

Response 12. This section was removed from the text on the suggestion of another reviewer.

Comment 13. Line 354: Human studies (e.g., Ulven et al., 2010) reported without mechanistic context. Suggestion adds, e.g., : ‘Null effects on urinary F2-isoprostanes [68] may reflect inadequate dosing or short duration for biomarker turnover.’

Response 13. Thank you for pointing this out. We agree with this comment. We added in the text.

Comment 14. Line 369: In Table 1, there is no need to mention the year of publication with the authors’ names. The reference number is sufficient.

Response 14. Thank you for pointing this out. We revised in the Table 1.

Comment 15. Line 427: Add references.

Response 15.  Thank you for your attention. We added it in the text.

Comment 16. Line 429-430: Rephrase to: ‘Conjugated double bonds in carotenoids (e.g., astaxanthin) dissipate singlet oxygen energy as heat, terminating oxidation cascades [83].’

Line 430-432: Rephrase to: ‘Astaxanthin’s polyene chain quenches lipid peroxyl radicals within membranes, while its polar end groups scavenge aqueous radicals at membrane interfaces [83, 84]’

Line 444-445: Please revise (NRF2 regulates AST), ‘NRF2 induces antioxidant enzymes (HO-1, NQO1)’

Response 16. Thank you for pointing this out. We revised all in the text.

Comment 17.  Line 460: The conclusion does not revisit the molecular focus promised in the Introduction. Reframe opening: ‘This review elucidates how compositional differences (e.g., phospholipids vs. triglycerides, astaxanthin vs. tocopherols) dictate distinct molecular actions.’

Response 17. Thank you for your suggestion. We added it in the text.

Comment 18. Line 463, 468: Tables and figures should not be referenced in the Conclusion section. Relevant sentences should be rephrased.

Response 17. Thank you for your suggestion. We fixed it in the text.

Line 468: I suggest adding: ‘Deciphering KO/FO pathway-specific efficacy will advance precision nutraceutical applications for inflammatory and oxidative pathologies. Future studies should quantify dose- and tissue-dependent differences in KO/FO pathway modulation (e.g., NRF2 activation kinetics, resolvin synthesis) to optimize clinical applications.’

Response 18. Thank you for your suggestion. We added it in the text.

Round 2

Reviewer 2 Report

Comments and Suggestions for Authors

Accept in present form.

Author Response

Dear Reviewer 2,

Thank you for your suggestions and contributions.

Kind regards,

Reviewer 3 Report

Comments and Suggestions for Authors

Congratulations to the authors! The efforts quite improved this valuable manuscript. It's scientifically sound but I would add some minor corrections to make sure it thoroughly follows the rules:
Please, make sure if Figure 1 should follow the sentence where it is first mentioned (Line 40), according to journal's rules.
Please, make sure that referencing authors throught the manuscript (including in Table 1) is uniform according to: Surname et al., year.
Attention to the references format. Please verify throughout the list if all the entered references are in accordance to the Journals rules.

Comments on the Quality of English Language

I would suggest just reviewing the overall manuscript for possible rephrasing regarding some long sentences.

Author Response

Dear Reviewer 3,

Thank you for your suggestions and contributions. The explanation of the corrections you requested is provided below and is highlighted in red in the text file.

Comment 1. Please, make sure if Figure 1 should follow the sentence where it is first mentioned (Line 40), according to journal's rules.

Response 1.  Thank you for pointing this out. We have revised.

Comment 2. Please, make sure that referencing authors throught the manuscript (including in Table 1) is uniform according to: Surname et al., year.

Response 2. Thank you for pointing this out. We deleted it upon the suggestion of another reviewer and added it again.

Commet 3. Attention to the references format. Please verify throughout the list if all the entered references are in accordance to the Journals rules.

Response 3.  Thank you for pointing this out. We have revised.

The article has been edited for language, and a certificate is presented.

Kind regards,